# An Adaptive Prompt Generation Framework for Task-oriented Dialogue System

**Jun Gao**[1], **Liuyu Xiang**[1], **Huijia Wu**[1], **Han Zhao**[2], **Yiqi Tong**[3], **Zhaofeng He**[1*]

[1]School of Artificial Intelligence, Beijing University of Posts and Telecommunications

[2] Didi chuxing

[3]School of Computer Science and Engineering, Beihang University

{jungao,xiangly,huijiawu,zhaofenghe}@bupt.edu.cn

zhaohan@didiglobal.com

yqtong@buaa.edu.cn

## Abstract

The de facto way of utilizing black-box large language models (LLMs) to perform various downstream tasks is prompting. However, obtaining suitable prompts for specific tasks is still a challenging problem. While existing LLM-based methods demonstrate promising performance in the task-oriented dialogue (TOD) task, they often require manual adjustment in prompt selection or focus solely on dialogue understanding or generation. To address these issues, we propose an adaptive prompt generation framework to fully unleash the potential of LLMs for the comprehensive TOD system. Firstly, we design a trainable slot generator (TSG) that can generate domain and slot information in the belief state, which serves as prior knowledge for subsequent prompt generation. Next, we propose an adaptive prompt generator (APG) that utilizes the prior knowledge to generate prompts for the LLM, deriving the belief state and system response of the dialogue for evaluation. Finally, we evaluate our framework on the MultiWOZ 2.0 dataset. Extensive experiments demonstrate that our method outperforms existing methods. Our code and data will be released.

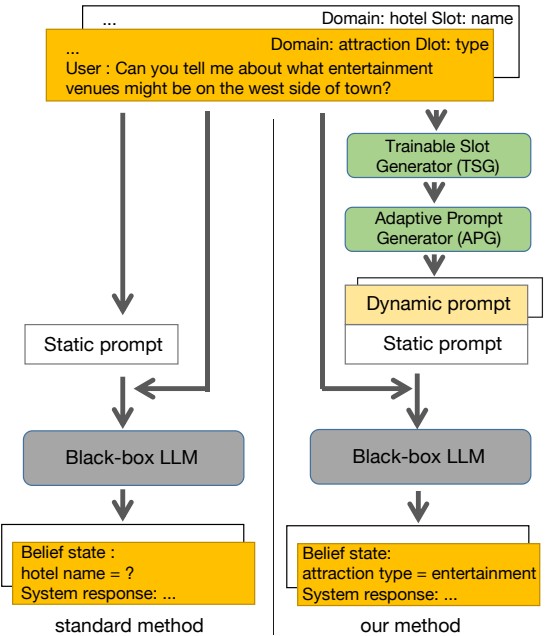

Figure 1: Comparison of our proposed Adaptive Prompt Framework with the standard prompting method for black-box LLM-based TOD system. Our method uses a trainable slot generator to obtain domain and slot, which serve as prior knowledge for prompt generation. This allows for better adaption to diverse dialogues involving various domains and slots.

## 1 Introduction

In recent years, significant progress has been made in LLMs, such as Instruct-GPT (Ouyang et al., 2022) and GPT4, and remarkable results have been achieved in their application to various downstream tasks such as task-oriented dialogue (TOD) and text summarization (Bang et al., 2023). Prompting has become a de facto method for utilizing black-box LLMs, as appropriate prompts can significantly enhance the capabilities of these models. However, different tasks require different prompts, and obtaining these prompts often requires manual adjustment. This is particularly challenging in dynamic scenarios like the TOD task, where prompt selection should adapt, usually indicating a large amount of manual labor is required.

In the context of the TOD system, two crucial components for measuring the success of a dialogue are belief state and system response. When using prompts to generate the system response from black-box LLMs, a problem arises: belief states and system responses vary with the domains and slots involved in dialogues. If the same prompt is used for all dialogues, the prompt becomes overly long and complex, resulting in hallucinations in LLMs. On the other hand, manually designing different prompts for different dialogues would incur a significantly higher labor cost and would be in-

---

*  Corresponding author.

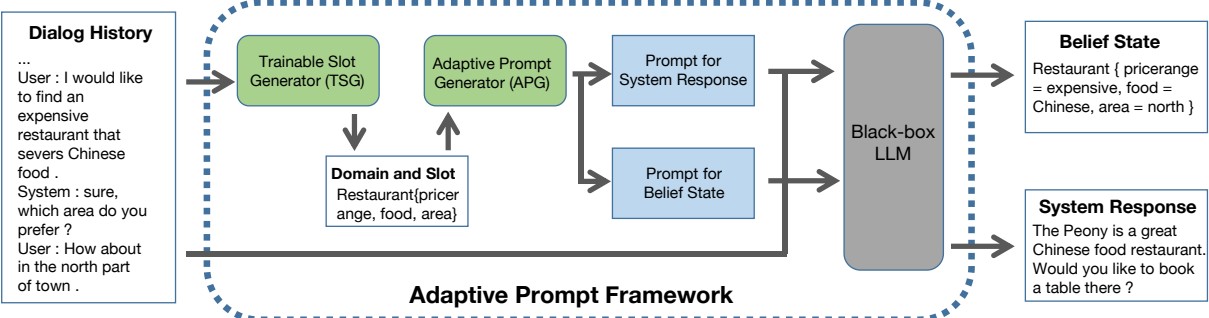

Figure 2: Overview of our proposed Adaptive Prompt Generation Framework.

convenient when extending the approach to new domains.

To address these issues, an automated method is needed to generate appropriate prompts for TOD tasks. One approach is the continuous prompt learning method, where prompts are represented as trainable vectors that can be concatenated with inputs (Shin et al., 2020; Zhang et al., 2022, 2023a; Swamy et al., 2023), enabling the acquisition of prompts suitable for the current input through gradient backpropagation. However, this approach lacks interpretability and requires model fine-tuning, making it unsuitable for black-box models (Diao et al., 2023). Another approach is to represent prompts using discrete tokens, which overcomes the limitations of vector-based prompts (Shin et al., 2020). However, this approach requires designing suitable prompts for different scenarios and these prompts often cannot be learned through training. As a result, it may lead to the suboptimal performance of LLMs and hallucinatory outputs (Bang et al., 2023).

Traditional methods rely on static or heuristic rules to construct prompts, but such methods are only suitable for simple scenarios. Recent approaches have achieved promising results in prompt generation. Sun et al. (2022) propose a black-box tuning framework but only applicable to continuous prompts. Diao et al. (2023) use policy gradient algorithm to find the optimal prompts, but only focus on classification tasks Li et al. (2023) propose a framework to provide prompt guidance for black-box frozen LLMs, but only focus on dialog response generation. Zhang et al. (2023b) propose a schema-guided prompting method for the TOD system, which still requires manual prompt design. Hence, there is currently no cost-effective method available to address the problem of adaptive prompt generation for black-box LLM-based

TOD systems.

In this paper, we propose an adaptive prompt generation framework for the comprehensive black-box LLM-based TOD system to address the aforementioned challenges. To obtain appropriate prompts with minimal data, we first extract domain and slot information from the belief state as training data and introduce a trainable slot generator (TSG) that can generate domains and slots involved in each dialogue turn. This approach reduces the annotation cost when expanding to other domains. We then design a generator (APG) using this information to generate domains and slots as prior knowledge for subsequent prompt generation. To automatically generate prompts suitable for the current dialogue state, we maintain a query table of prior knowledge and candidate values. We utilize the previously generated prior knowledge to select suitable entries from the list and compose prompts that capture the current dialogue belief state and system response. This way, the content generated by the LLM is precisely constrained by tailored prompts, which include the desired candidates without introducing redundant content.

The main contributions of our work can be summarized as follows:

1. We design a prompt construction method based on domain and slot information.

2. We proposed an adaptive prompt generation framework for the comprehensive black-box LLM-based TOD system.

3. Experimental results demonstrate the effectiveness of our approach in enhancing the capabilities of LLMs.

## 2 Related work

### 2.1 Black-Box LLMs in Downstream Tasks

In recent years, there has been a proliferation of large language models such as Codex (Chen et al.,

2021), LaMDA (Thoppilan et al., 2022), PaLM (Chowdhery et al., 2022), chatGPT, GPT4, etc., which have greatly enhanced various downstream tasks in NLP. However, most of these LLMs are not open source and can only be accessed through query and prediction interface (Diao et al., 2023). As a result, a multitude of studies have emerged that focus on prompting methods tailored to specific tasks for black-box LLMs. (Sun et al., 2022) propose a black-box tuning framework but only applicable to common language understanding tasks. (Diao et al., 2023) use policy gradient algorithm to find the optimal prompts, but only focus on classification tasks. (Li et al., 2023) propose a framework to provide prompt guidance for black-box frozen LLMs, but only focus on dialog response generation. (Pan et al., 2023) specifically focuses on optimizing prompts for the dialogue understanding task. In summary, they are not applicable to our task. In our work, we propose an adaptive prompt generation framework for the comprehensive TOD system.

## 2.2 Prompt Learning

Another line of our work involves prompt learning which finds optimal prompts suitable for specific tasks. One common approach is to train continuous prompts (Liu et al., 2021; Lester et al., 2021; Gu et al., 2022). However, these prompt types are not easily interpretable and require LLMs for training or fine-tuning. Consequently, some studies have proposed constructing prompts in a discrete manner (Shin et al., 2020; Gao et al., 2021; Sun et al., 2022), using generated or manually crafted prompts. (Schick and Schütze, 2021) propose the pattern-verbalizer pair (PVP) method, which constructs prompts by selecting appropriate patterns and verbalizers. Inspired by the aforementioned work, we propose a novel approach to generate appropriate prompts for the comprehensive TOD system.

## 3 Method

Due to the phenomenon of hallucination that occurs during the application of black-box LLMs, it is hard to generate belief states accurately required for TOD tasks. Moreover, as the prompt becomes more complex, the content generated by black-box LLMs, like ChatGPT, becomes even more uncontrollable. Therefore, a method is needed to constrain the content generated by ChatGPT.

| Domain | Attraction | Train | Hotel | Restaurant |
|--------|-----------|-------|-------|-----------|
| Train  | 50        | 50    | 50    | 50        |
| Valid  | 50        | 50    | 50    | 50        |
| Test   | 100       | 200   | 200   | 200       |

Table 1: Data statistics of four single-domain dialog datasets (Budzianowski et al., 2020; Peng et al., 2021)

### 3.1 Data Preparation

In this section, we describe the steps involved in preparing the data for our experiments. Our approach involves selecting a small subset of belief states from different domains in the MultiWOZ 2.0 dataset, based on the settings of SOLOIST. Subsequently, the selected data is processed to remove the value component.

**Selection of Belief State Data Subset:** We follow the setting of SOLOIST (Peng et al., 2021) and choose a limited number of dialogues that contain only one domain from the MultiWOZ 2.0 dataset. The data statistics of each domain are shown in Table1.

**Processing of Selected Data:** To train a generator that only generates domains and slots of the belief state, we perform further processing to eliminate the value component from the belief state. This step involves removing the actual values associated with each slot in the belief state while retaining the slots and domains. By removing the value component, we focus solely on the task of predicting the belief state without considering specific slot values.

By following these steps, we prepare the dataset for our experiments, which enables us to train and evaluate TSG.

### 3.2 Trainable Slot Generator

In Dialogue State Tracking (DST) task, We use SOLOIST with pre-trained weights and then fine-tuned on a small amount of data obtained in Section 3.1. Nevertheless, unlike SOLOIST, our model is fine-tuned to generate domains and slots of belief states only. In this process, the TSG will generate prior knowledge which can be used to generate prompts in subsequent adaptive prompt generation framework.

Specifically, we can represent each dialog turn in the training dataset as :

$$x = (s, b)$$

where $s$ is the dialog history up to the current dialog turn, $b$ is the annotated belief state only with domain and slots.

In the TOD system, the joint probability $p(x)$ can be expressed in the form of an autogressive model:

$$p(x) = p(b, s)$$
$$= p(b \mid s)p(s)$$

where $p(b \mid s)$ refers to predicting the slots in the belief state. Note that we solely focus on predicting the domains and slots of belief states, excluding the values of belief state and system response, as they are generated in the subsequent part.

If we define the length of the belief state, which consists solely of domains and slots, as $T$, the training objective for this process can be denoted as:

$$\mathcal{L}_1 = \log p(b \mid s) = \sum_{t=1}^{T} \log p_\theta \left( b_t \mid b_{<t}, s \right)$$

where $\theta$ represents the model parameters to be learned.

In line with the approach of SOLOIST, we have also incorporated a contrastive learning training objective to improve the efficiency of our model's learning process. However, unlike SOLOIST, when constructing negative examples (represented as $x'$), we do not perform a complete replacement of the entire belief state. Instead, we perform separate replacements on the domain and slots components within the altered belief state $b$. We can define the label for a belief match as $y = 1$ and the label for a non-match as $y = 0$. In this case, the training objective of contrastive learning can be formulated as:

$$\mathcal{L}_2 = y \log \left( p_\theta(x) \right) + (1 - y) \log \left( 1 - p_\theta \left( x' \right) \right)$$

The three types of negative samples generated by our approach are as follows:

- Only replacing the domain component of the modified belief state.

- Only replacing the slots component of the modified belief state.

- Simultaneously replacing both the domain and slots components of the modified belief state.

Finally, the proposed method in this section is model-agnostic and can be applied with other models interchangeably. However, due to code and data availability, we have only validated the approach using the SOLOIST model in this paper.

## 3.3 Adaptive Prompt Generation Framework for TOD System

In the TOD task, the belief state serves as an indication of the user's intent and also acts as an intermediate state for extracting external information. Existing methods have demonstrated that LLMs perform well in acquiring the belief state. However, they also face certain challenges. One important factor is that large-scale models are sensitive to prompts. Clear and concise prompts tend to yield better results, while vague and lengthy prompts may lead to unexpected outcomes.

The TOD task differs from traditional tasks in that the domain and intent of user utterances change in each dialogue turn. When extracting information from such dialogues using LLMs, using the same prompt may not effectively capture the required belief state information. On the other hand, using different prompts for each dialogue can significantly increase manual effort.

To address this, we propose a model-agnostic adaptive prompt generation framework in this section. This framework assists LLMs in generating the belief state and system response of dialogues. The overall structure of the framework is illustrated in Figure 2.

**Adaptive Prompt Generator (APG):** The main function of this generator is to generate prompts that are required to obtain the final belief state and system response. Since the prompts dynamically change based on the dialogue process, we utilize the partial belief state generated in Section 3.2. In addition to this, the candidate lists of domains and slots $L_{ds}$ are required to obtain the complete belief state, as well as the special token lists $L_{st}$ are required to generate the system response. Next, it is necessary to parse the generated partial belief state $b$ and generate domain $s_d$ and slot information $s_s$. We can consider the prompts input to the LLM as a function related to the partial belief state $b$. The prompts for generating the complete belief state and system response can be represented as $f_1$ and $f_2$, respectively.

**Prompt for Belief State:** The prompt for generating the complete belief state consists of two parts: static and dynamic. The static part includes standard prompt and data examples, partly referenced from (Bang et al., 2023), and can be represented as $Prompt_{s1}$. The dynamic part involves domain slots and candidates matched by $s$ and can be de-

fined as $Prompt_{d1}$. The candidates of slots can be divided into two categories: one category includes slots with limited candidates that can be obtained from the dataset, such as the '*area*' slot with candidates *['centre', 'east', 'north', 'south', 'west']*. The other category includes candidates that cannot be exhaustively listed, such as the name slot representing various entity names related to different domains in user intents. The prompt function for generating the complete belief state utilizes the previously partial belief state containing only the domain and slot information, and can be represented as follows:

$$f_1(\boldsymbol{b}) = Prompt_{s1} + Prompt_{d1}(\boldsymbol{b}, L_{ds})$$

where specific $\boldsymbol{b}$ has a fixed mapping relationship with the elements in $L_{ds}$.

Once we have the partial belief state $\boldsymbol{b}$ containing only the domain and slot information, we can use its prompt function $f_1$ to generate the prompt. This prompt is then inputted into the LLM, $p_{LLM}$, to obtain the final result:

$$\boldsymbol{s}^* = p_{LLM}(f_1(\boldsymbol{b}), \boldsymbol{s})$$

where $\boldsymbol{s}^*$ represents the complete belief state in the current dialogue state.

**Prompt for System Response:** The prompt for generating the system response also consists of static and dynamic components. The static part serves as a guidance for the LLM to simulate the generation of system response and can be represented as $Prompt_{s2}$. The dynamic part involves matching special tokens based on $\boldsymbol{s}_d$, which are placeholders used to represent system-recommended entities. The correct selection of these special tokens is crucial for evaluating the success of the dialogue and can be represented as $Prompt_{d2}$. The prompt function for generating the system response requires the dialogue history $s$ and can be represented as:

$$f_2(\boldsymbol{s}_d) = Prompt_{s2} + Prompt_{d2}(\boldsymbol{s}_d, L_{st})$$

where specific $\boldsymbol{s}_d$ has a fixed mapping relationship with the elements in $L_{st}$, and both $f_1$, $Prompt_{s1}$, $Prompt_{d1}$, $f_2$, $Prompt_{s2}$, $Prompt_{d2}$ are all in string format, and the "+" operator denotes direct string concatenation.

Next, we need to obtain the system response using the LLM. Based on the previously obtained prompt function, the final result can be represented as:

$$r = p_{LLM}(f_2(\boldsymbol{s}), \boldsymbol{s})$$

where $r$ represents the delexicalized system response.

Therefore, the combined output of the belief state and system response generated by the LLM for evaluation is:

$$y = p_{LLM}(f_1(\boldsymbol{b}), \boldsymbol{s}) + p_{LLM}(f_2(\boldsymbol{s}), \boldsymbol{s}).$$

## 4 Experiment

### 4.1 Dataset and Evaluation

We evaluated the effectiveness of the proposed method on the MultiWOZ single-domain dialog datasets (Budzianowski et al., 2020), reorganized by (Peng et al., 2021). This dataset consists of four domains: Attraction, Hotel, Restaurant, and Train.

In the TSG component, we used the standard metric in DST: joint goal accuracy (JGA). This metric compares the predicted belief state with the ground truth belief state to determine if they are completely identical. A successful prediction is achieved when the predicted belief state matches the ground truth belief state entirely. Note that the JGA metric requires both value and slot information to match for a successful evaluation. In our work, the trainable slot generator only generates domain and slot information of the belief state and does not include value information. Therefore, the evaluation is performed by complementing the value information using the LLM in the second part before conducting the evaluation.

For the evaluation of the entire dialogue, we concatenate the belief state and system response generated by the LLM in the order of the dialogue and then evaluate them. We employ the following evaluation metrics: Inform, which measures whether the provided entities satisfy the user's needs correctly; Success, which measures whether all requested attributes are addressed; BLEU is not adopted because it measures the similarity between the generated response and the reference response, and there may be significant differences between the responses generated by the LLM and the reference responses.

### 4.2 Implementation Details

The slot generator and adaptive prompt framework proposed in our approach are not restricted to specific models. In this paper, the TSG is based on

GPT-2 with 117M parameters and initialized with pretraining weights from SOLOIST. SOLOIST is pre-trained on corpora from multiple task-oriented dialogue datasets (Schema and Taskmaster) and performs tasks such as belief state prediction and response generation. Specifically, the input for the belief state prediction task is the current dialogue and all dialogue history, and the output is a text sequence corresponding to the current dialogue state. In our task, we only need the domain and slot information from the belief state. An intuitive approach would be to directly extract domain and slot information from the belief state generated by the original model. However, this method would result in more complex and longer text sequences, requiring more time but not improving accuracy. Therefore, we only trained the model using the domain and slot parts of the belief state.

In the experimental part of the adaptive prompt framework, the LLM we use is chatgpt-3.5-turbo. We generated belief states and system responses of dialogues by calling OpenAI's API. We find the format of the generated content can affect its accuracy, if we use curly braces to restrict the belief state, such as "*{belief state: attraction type = entertainment}*", the result is better than when not using curly braces, like "*belief state: attraction type = entertainment*". Based on our experience, we control the format of the generated content to achieve optimal results. Therefore, belief states and system responses generated during the intermediate process may differ slightly in format from those in SOLOIST.

### 4.3 Trainable Slot Generator (TSG)

#### 4.3.1 Setup

The goal of the TSG is to generate belief states that only contain slot information. We use SOLOIST as a baseline to compare the improvement of our method on the final results. First, we train the TSG following SOLOIST, and the data statistics of dataset are shown in Table 1. The training epoch is 30, the learning rate is 5e-5, and the batch size is 1. During testing, Nucleus filtering is used for decoding with a top-p value of 0.4 and a temperature of 0.3 to obtain the belief state. Since we only need the slot part of the belief state, we parsed the belief state and generated a new belief state that only includes the domain and slot information using rules-based methods.

For our proposed method, we only used the do-

main and slot information from the belief state instead of the complete belief state for training. This approach has two advantages as follows:

- In case manual annotation of data is required for future applications, annotators only need to label the domain and slot information of dialogues. Unlike value information, which is diverse and difficult to standardize, these two types of information are usually easier to determine and have a certain range. This significantly reduces the annotation cost.

- Training and inferring with belief states that contain only domain and slot information make the input and output sequences of the model shorter. Compared to complete belief states, which contain more characters representing the format and content, this method achieves better stability in generating results.

| | Attraction | Hotel | Restaurant | Train |
|---|---|---|---|---|
| ChatGPT | 58.80 | 30.76 | 45.31 | 59.76 |
| PPTOD | 57.71 | 31.89 | 54.12 | 61.78 |
| PPTOD* | 58.11 | 32.76 | 54.34 | 62.90 |
| SOLOIST | **60.10** | 27.83 | 54.94 | 63.30 |
| TSG w/o CL | 58.11 | 32.06 | 55.21 | 63.25 |
| TSG | 58.27 | **33.02** | **55.81** | **63.78** |

Table 2: JGA results across different methods and domains. For a fair comparison, JGA is evaluated only based on the domain and slot, excluding the consideration of values. "*" indicates the replacement of corresponding sections with the TSG method.

| | Inform | Success | BLEU | Combined score |
|---|---|---|---|---|
| SOLOIST | 73.88 | 72.22 | 13.11 | 86.16 |
| SOLOIST+TSG | **76.23** | **74.10** | **13.81** | **88.98** |

Table 3: Result on Camrest676 dataset.

#### 4.3.2 Results

The results obtained by different methods are shown in Table 2. We observe that directly utilizing ChatGPT for extracting slot information of belief state is not particularly effective. The method trained with shorter belief states surpasses the results obtained by extracting slot information from the original SOLOIST method, and is also better than PPTOD and ChatGPT. We can observe that in more complex domains, such as hotels, the improvement becomes more pronounced as the belief state becomes more complex. This indicates that using less information for training can lead to more

accurate results, validating the potential of our proposed method. It is worth noting that the JGA metric of belief states containing only domain and slot information reflects the algorithm's advantage from one aspect. Further application in subsequent methods is needed to verify its actual improvement in the final dialogue evaluation.

To assess the generalizability of our proposed method, we applied the TSG to the Camrest676 dataset. The results as shown in Table 3 indicate that, even when tested on a novel dataset, our proposed method consistently enhances the results. This observation underscores the method's capacity to generalize to datasets beyond its training domain.

### 4.4 Adaptive Prompt Framework

#### 4.4.1 Setup

The purpose of the Adaptive Prompt Framework is to generate suitable prompts for inputting the current dialogue into the LLM to obtain the corresponding belief state and system response. The underlying assumption is that the LLM possesses strong comprehension and generalization capabilities and can adapt to various downstream tasks by designing appropriate prompts. Specifically, the Adaptive Prompt Framework consists of two parts:

Part 1 involves refining the results from the TSG. Firstly, a candidate list of domains, slots, and their corresponding values is compiled based on the dataset. This list serves the purpose of retrieving the candidate values for a given domain and slot when we obtain the domain and slot information. This candidate list includes two types of entries. The first type comprises slots with limited values, where the values can be represented by short-word sequences, such as slots indicating time or direction. The second type consists of slots with potentially infinite values, where the values may vary during the conversation, such as the "$name$" slot that may change with different entities discussed in the dialogue. These entries cannot be exhaustively enumerated, and thus we use "?" to represent candidate values. Each entry in the candidate list pertains to a single domain and slot but can contain multiple candidate values.

Next, we incorporate the static prompt section, based on existing work on generating prompts for belief states. However, these prompts are not adaptable to subsequent dynamic utterances. Therefore, we adjust them by adding instructions on how to

use the subsequent dynamic section and include an example to provide constraints on the generated format.

Then, we parse the results generated by the TSG, extract the domain and slot information, and select the relevant entries from the existing domain-slot-value candidate list to form a list of candidate entries.

Finally, we process the input dialogue information to generate the second dynamic part of the prompt. This part consists of current user utterances, dialogue history, and partial belief state, combined into a string list format. The partial belief state is represented as "$\{slot\ =?\}$". The example prompt is presented in Appendix A.

Part 2 involves generating the system response based on the user utterance and dialogue history in the current conversation. The purpose of this part is to have the LLM act as an agent in the TOD system and respond to user requests by generating delexicalized system responses. It is important to note that, for the convenience of evaluating the success rate of subsequent dialogue, recommended or queried entities in the generated response need to be replaced with special tokens. The complete response is then generated through post-processing.

Similar to the previous part, we need to compile a list of domains and corresponding special tokens based on the dataset. This list serves the purpose of retrieving candidate values for a given domain using table lookup when we obtain the domain information, to provide the LLM with options for generating the final system response. This candidate list includes two types of special tokens. The first type consists of domain-specific special tokens represented in the form of domain-slot, such as "attraction-name". The second type comprises special tokens that are universally applicable across all domains and are represented as value-slot, such as "value-time". These special tokens represent some commonly occurring entities that may be encountered in all domains.

Next, we incorporate the static prompt section for generating the system response based on the belief state prompt generated in the previous part.

Then, we generate the second dynamic section of the system response prompt using the current user utterance and dialogue history.

Finally, based on the parsing results from the slot generator, we obtain the domain information and select the relevant entries from the domain-

| | Attraction | | | Hotel | | | Restaurant | | | Train | | |
|---|---|---|---|---|---|---|---|---|---|---|---|---|
| | dst | inform | success | dst | inform | success | dst | inform | success | dst | inform | success |
| SOLOIST | / | 86.00 | 68.00 | / | 75.00 | 51.50 | / | 84.00 | 62.5 | / | 81.30 | 74.20 |
| GALAXY | / | 92.00 | 62.00 | / | 84.50 | 29.00 | / | 76.50 | 64.50 | / | 87.31 | 73.60 |
| ChatGPT | 59.98 | 95.00 | 86.00 | 28.30 | 89.50 | 43.00 | 53.98 | 95.00 | 61.50 | 59.72 | 83.70 | 77.70 |
| ours w/o APG | 45.14 | 84.00 | 73.00 | 8.50 | 48.50 | 36.50 | 15.09 | 61.50 | 46.50 | 2.06 | 81.22 | 29.95 |
| ours w/o TSG | 50.13 | 89.00 | 78.00 | 24.06 | 77.00 | 37.50 | 47.56 | 86.50 | 57.50 | 51.82 | 81.73 | 75.63 |
| ours | 62.20 | 98.00 | 87.00 | 28.90 | 88.50 | 62.00 | 54.90 | 96.50 | 71.50 | 61.70 | 82.74 | 80.71 |

Table 4: Results on four tasks. SOLOIST is quoted from (Peng et al., 2021) and we use the gpt-3.5-turbo-0301 version of ChatGPT.

| | Attraction | | Hotel | | Restaurant | | Train | |
|---|---|---|---|---|---|---|---|---|
| | BLEU | Combined Score | BLEU | Combined Score | BLEU | Combined Score | BLEU | Combined Score |
| SOLOIST | 14.60 | 91.60 | 10.09 | 73.34 | 13.17 | 86.42 | 11.90 | **89.18** |
| GALAXY | 9.47 | 86.47 | 5.50 | 62.25 | 11.68 | 82.18 | 6.67 | 87.13 |
| ChatGPT | 4.11 | 94.61 | 2.12 | 68.37 | 3.20 | 81.45 | 3.56 | 84.26 |
| ours | 4.89 | **97.39** | 2.76 | **78.01** | 3.51 | **87.51** | 4.98 | 86.71 |

Table 5: Results of combined score on four tasks. Calculated from BLEU, Inform, and Success metrics in Table 4 using the formula $Combined\ Score = (Inform + Success)/2 + BLEU$.

special token candidate list to form a list of candidate entries. The example prompt is presented in Appendix A.

### 4.4.2 Compared Methods

To demonstrate the effectiveness of our proposed method, we compared it with several different combinations of methods for validation. The main methods we employed are as follows:

- SOLOIST: Using SOLOIST alone to generate belief states and system responses, followed by direct evaluation.

- ChatGPT: Generating belief states and system responses separately using ChatGPT. Then, the two parts are concatenated for evaluation. In contrast to our proposed method in this section, this approach uses static prompts. Specifically, when generating belief states, the dynamic prompt section from our proposed method, which consists of the candidate list based on the output of TSG, is replaced with the complete domain-slot-value list. Similarly, when generating system responses, the dynamic prompt part from our proposed method, which consists of the candidate list based on the domain information, is replaced with the complete domain-special token list. In other words, this method solely relies on ChatGPT to generate the required information, aiming to verify if ChatGPT is capable of completing the entire task.

- ours w/o APG: The domain and slot information is generated using TSG, and then the static prompt is used to input into ChatGPT for generating the final belief state and system response.

- ours w/o TSG: Using SOLOIST to generate belief states. After that, the prompt is generated using APG and input into ChatGPT to generate system responses.

These methods were evaluated by concatenating the corresponding components and assessing their performance.

### 4.4.3 Results

The final evaluation results are presented in Table 4. We use SOLOIST as the baseline. It can be observed that the ours w/o TSG method achieves an average improvement of 5 points. This improvement can be attributed to the powerful comprehension and generation capabilities of ChatGPT, allowing it to generate responses that align well with the conversational context. The ChatGPT method alone brings an average improvement of 1 point, indicating its capability to generate belief states effectively. However, we noticed that ChatGPT performs better in generating simple belief states (e.g., in the attraction domain) but struggles with complex belief states (e.g., in the train domain). We speculate that this discrepancy is due to complex belief states containing more information, and ChatGPT's instability in generating longer pieces of information. Ultimately, our proposed method

| | Attraction | | | | Hotel | | | | Restaurant | | | | Train | | | |
|---|---|---|---|---|---|---|---|---|---|---|---|---|---|---|---|---|
| | Info. | Succ. | BLEU | Comb. | Info. | Succ. | BLEU | Comb. | Info. | Succ. | BLEU | Comb. | Info. | Succ. | BLEU | Comb. |
| ours- | 85.00 | 69.00 | 12.90 | 89.90 | 74.50 | 43.50 | 8.12 | 67.12 | 81.00 | 55.50 | 12.80 | 81.05 | 80.81 | 64.65 | 9.96 | 82.69 |
| ours | 98.00 | 87.00 | 4.89 | 97.39 | 88.50 | 62.00 | 2.76 | 78.01 | 96.50 | 71.50 | 3.51 | 87.51 | 82.74 | 80.71 | 4.98 | 86.71 |

Table 6: Experimental results on the impact of static and dynamic prompts on system response. "-" indicates the substitution of the dynamic prompt segment with a static prompt encompassing all entries.

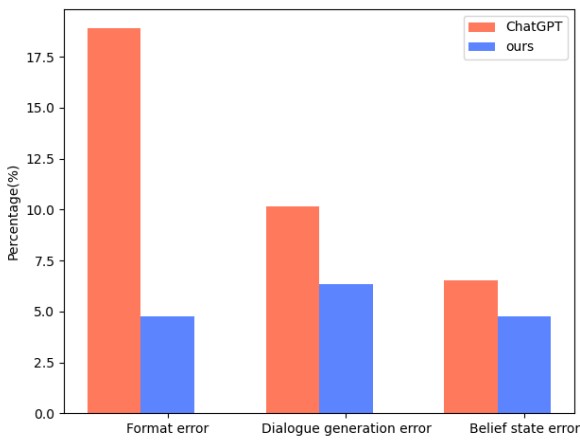

Figure 3: Comparison of different methods in terms of the proportion of high-frequency errors relative to the total test data.

outperforms the ChatGPT method with an average improvement of 8 points. This indicates our framework effectively mitigates the issue of hallucination in ChatGPT.

## 5   Analysis

**High-frequency Errors.** For the unsuccess dialogues, we analyze high-frequency errors comparing the chatgpt method in the restaurant domain with our proposed method, as shown in Figure 3. Specifically, "format error" is the error that prevents the belief state from being parsed, "dialogue generation error" refers to the result of a system response that does not align with logic, and "belief state error" refers to unmatched content.

Our method significantly reduces the number of evaluation failures caused by format errors owing to the inclusion of precise constraints in the prompt. Furthermore, the proportion of dialogue generation errors is also reduced. Lastly, the occurrence of belief states' errors has been mitigated due to the trainable slot generator.

**Drawback of Static Prompt.** While the present study has already included a comparison between static and dynamic prompts, to more distinctly illustrate the differences between existing methods and our proposed approach, we conducted a com-

| | Attraction | Hotel | Restaurant | Train |
|---|---|---|---|---|
| Bang et al. 2023 | 91.00 | 83.50 | 90.50 | 77.27 |
| Pan et al. 2023 | 93.00 | 84.00 | 91.00 | 78.82 |
| ours | **98.00** | **88.50** | **96.50** | **82.74** |

Table 7: Comparison of results with different prompts.

parative analysis of the effects of prompts using the frameworks presented in Bang et al. (2023) and Pan et al. (2023).

As depicted in Table 7, the dynamic prompt exhibits a significantly higher performance in the inform metric compared to existing static prompt methods. This is attributed to the inability of static prompts to adapt to different domains.

Similarly, to investigate the impact of static and dynamic prompts on system response, we supplemented our study with more detailed experiments to compare their effects on the final results. Specifically, we replaced the dynamic part of the prompt with a static prompt containing all entries. As shown in Table n, using only a static prompt has a significant impact on success, resulting in a decrease in the combined score.

**Necessity of Belief State in TOD.** Due to the inability of LLMs to interact extensively with external knowledge, such as querying restaurant availability from a DB, we still need to retrieve keywords from the belief state to perform DB queries. While there may be better approaches in the future, currently, querying external knowledge through the belief state remains a more reliable method.

## 6   Conclusion

To address the issue of prompt generation of LLMs in the TOD task, we propose an adaptive prompt generation framework for the comprehensive TOD system which consists of two parts: trainable slot generator (TSG) and adaptive prompt generator (APG). The framework tackles the limitation of fixed prompts in TOD and focuses on both dialogue understanding and dialogue generation. Experimental results demonstrate that our proposed method significantly improves the performance compared to existing approaches.

## Limitations

Limitations There are several aspects of this article that can be improved in the future:

**Result Updates:** As ChatGPT continues to evolve, we can continue using newer versions to enhance performance.

**More Scenarios:** Currently, we have only experimented with single-domain task-oriented dialogues. In the future, we can improve the model's performance in multi-domain scenarios or incorporate additional information, such as database query information, into the prompts.

**Generalization validation:** In the future, we can expand our approach to compare it with other methods in a broader range of tasks and validate its generalization capabilities.

## Acknowledgements

The authors would like to thank Xiaoying Zhang for useful discussions. This work is supported by National Key R&D Program of China (Grant No.2022YFF1202400), Major Technology Innovation Program of Hangzhou, China (Grant 2022AIZD0154), National Natural Science Foundation of China (No.62176025), National Natural Science Foundation of China (No.62301066), Beijing Nova Program 20220484161, and the Fundamental Research Funds for the Central Universities 2023RC72. This work is also sponsored by CCF-DiDi GAIA Collaborative Research Funds for Young Scholars.

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

# A Appendix

## A.1 Prompt for Belief State

The various parts of the prompt for obtaining the belief state are presented in Table 8. The dynamic part automatically adapts based on different inputs to generate the optimal prompt.

| Part | Prompt |
|------|--------|
| Input | (utterance)
user : I'd like to visit a college in the center of town. could you help me find something interesting? |
| Static | (example)
user : hello, i am looking for something to do in the west part of town. it should involve multiple sports.
system : unfortunately none of those places exist here . any other preferences ?
user : hm, can you tell me about what entertainment venues might be on the west side of town instead?
=>
{ belief : attraction type = ? }
=>
{ belief : attraction type = entertainment }

(standard prompt)
According to the example, fill the blank represented as '?' of the dialogue state of the last utterance in the following dialogue by using the following pre-defined slots and possible values: |
| Dynamic | (relevant entries)
Domain: attraction, type : ['dontcare', 'park', 'mutliple sports', 'boat', 'college', 'cinema', 'nightclub', 'concerthall', 'swimmingpool', 'museum', 'entertainment', 'theatre', 'architecture']
Domain: attraction, area : ['dontcare', 'centre', 'east', 'north', 'south', 'west']

(TSG output)
=>
{belief : attraction type = ? ; area = ? }
=> |
| Output | {belief : attraction type = college ; area = center } |

Table 8: Prompt for belief state

## A.2 Prompt for System Response

Similarly, the prompt for obtaining the system response is presented in Table 9. The dynamic part also selects relevant entries based on the output of TSG.

| Part | Content |
|------|---------|
| Input | (utterance)
user : I'd like to visit a college in the center of town. could you help me find something interesting? |
| Static | (example)
user : hello, i am looking for something to do in the west part of town. it should involve multiple sports.
system : unfortunately none of those places exist here . any other preferences ?
user : hm, can you tell me about what entertainment venues might be on the west side of town instead?
=>
system : there s a fun place called [attraction_name] at [attraction_address].

(standard prompt)
According to the example, complete the system without generating unnecessary elements. Special tokens in the system can be used: |
| Dynamic | (relevant entries)
'[attraction_address]', '[attraction_area]', '[attraction_name]', '[attraction_phone]', '[attraction_postcode]', '[attraction_pricerange]', '[attraction_reference]', '[value_count]', '[value_day]', '[value_place]', '[value_time]' |
| Output | system: sure, there are several colleges in the center of town. one of the more interesting ones is [attraction_name] located at [attraction_address]. |

Table 9: Prompt for system response