# OpenReview forum: "An Adaptive Prompt Generation Framework for Task-oriented Dialogue System"
_EMNLP/2023/Conference — EMNLP 2023 Findings_

### Official Review · Reviewer_RXui · 2023-07-25

**Soundness:** 2

**Excitement:**

2: Mediocre: This paper makes marginal contributions (vs non-contemporaneous work), so I would rather not see it in the conference.

**Paper Topic And Main Contributions:**

Large Language Models (LLMs) have demonstrated impressive performance in various tasks. However, it usually requires different prompt designs for different specific tasks. Therefore, in this paper, the authors propose an adaptive prompt-generation framework to automatically generate prompts for task-oriented dialogue. More specifically, the authors design a trainable slot generator (TSG) to generate domain and slot information in the belief state, and then propose an adaptive prompt generation (APG) to utilize the prior knowledge to generate prompts for LLM. Experimental results also demonstrate the proposed method.

**Questions For The Authors:**

1. Can you provide some descriptions of the belief state and what is its structure?
2. Are there any ablation studies to investigate the effect of static prompt, dynamic prompt and the prompt for system response to the final results?
3. How to measure the format error and dialogue generation error?

Another question about this area: This paper introduces an Adaptive Prompt Generation for solving task-oriented dialogue tasks, and it introduces many previous experiences (e.g., Belief state) in TOD tasks. But one question is: after we have LLMs (e.g., ChatGPT), do we still need to follow these traditional settings to solve TOD tasks?

**Reasons To Accept:**

1. This paper proposes an adaptive prompt generation framework to solve task-oriented dialogue tasks.

**Reasons To Reject:**

1. The proposed Trainable Slot Generator is only validated on the SOLOIST model, which is incremental and minor. Besides, why incorporating contrastive learning can improve efficiency should be explained.
2. It seems that the proposed TSG is tuned on a small amount of data, and I am worried about whether the proposed method can be generalized to other domains. The motivation of this paper is ok, but if the proposed method is trained on the MultiWOZ dataset and can only solve tasks in MultiWOZ, its value will be minor.
3. This paper argues that using the same prompt in LLM cannot effectively capture useful belief state information. Can you provide some experiments to validate this point? And whether the belief state is necessary in TOD tasks when we have LLMs?

**Reproducibility:**

2: Would be hard pressed to reproduce the results. The contribution depends on data that are simply not available outside the author's institution or consortium; not enough details are provided.

**Reviewer Confidence:**

4: Quite sure. I tried to check the important points carefully. It's unlikely, though conceivable, that I missed something that should affect my ratings.

---

> ### Author Rebuttal · Authors · 2023-08-28
>
> __Reviewer#3:__
>
> We appreciate your thoughtful and invaluable comments. We will explain your concerns point by point.
> ***
>
> __Q1:__ The proposed Trainable Slot Generator is only validated on the SOLOIST model, which is incremental and minor. Besides, why incorporating contrastive learning can improve efficiency should be explained.
>
> __A1:__
>
> **Effectiveness of TSG**
>
> It is worth noting that the proposed TSG can be combined with any other TOD model apart from SOLOIST. To verify its effectiveness, we replace SOLOIST with PPTOD, and the results below demonstrate the effectiveness of the proposed TSG.
>
> | |Attraction | Hotel | Restaurant | Train |
> |:------|:------:|:-------:|:------------:|:-------:|
> | PPTOD w/o TSG    | 57.71 | 31.89      | 54.12 | 61.78 |
> | __PPTOD w/ TSG__      | __58.11__ | __32.76__      | __54.34__ | __62.90__ |
>
> **Effectiveness of contrastive learning**
>
> Contrastive learning enhances training effectiveness and efficiency because it constructs negative domain and slot examples, allowing the model to learn from incorrect results, thus accelerating model convergence. The table below compares the original method with the one using contrastive learning.
>
> |                        | Attraction | Hotel | Restaurant | Train |
> |:--------------------------:|:------------:|:-------:|:----------:|:------:|
> | w/o Contrastive Learning | 58.18      | 32.06 | 55.21      | 63.25 |
> | __w/ Contrastive Learning__  | __58.27__      | __33.02__ | __55.81__      | __63.78__ |
>
> ***
> __Q2:__ It seems that the proposed TSG is tuned on a small amount of data, and I am worried about whether the proposed method can be generalized to other domains. The motivation of this paper is ok, but if the proposed method is trained on the MultiWOZ dataset and can only solve tasks in MultiWOZ, its value will be minor.
>
> __A2:__ In order to verify whether the proposed method can be generalized to other domains, we conduct experiments on Camrest676 dataset and the result is shown in the table below.  We choose SOLOIST as the baseline method. The result shows that the proposed TSG is not limited to MultiWOZ dataset, but can also bring consistent improvements on other datasets.
>
> | _Camrest676_ | Inform | Success | BLEU  | Combined score |
> |:----------|:------:|:-------:|:-----:|:--------------:|
> | w/o TSG    | 73.88  | 72.22   | 13.11 | 86.16          |
> | __w/ TSG__     | __76.23__  | __74.10__   | __13.81__ | __88.98__          |
>
> ***
> __Q3:__ This paper argues that using the same prompt in LLM cannot effectively capture useful belief state information. Can you provide some experiments to validate this point? And whether the belief state is necessary in TOD tasks when we have LLMs?
>
> __A3:__
>
> **Drawback of static prompt**
>
> In the original paper, 'ours w/o APG' refers to experimental results using a static prompt, which utilizes the static part from Appendix A.1 along with complete entries. In addition to this, we conducted several sets of experiments, as shown in the table below, to compare the results when using the same prompts, as referenced in the recent paper.
>
> | _Inform_ | Attraction | Hotel | Restaurant | Train |
> |:------:|:----------:|:-----:|:----------:|:-----:|
> | __ours__   | __98.00__      | __88.50__ | __96.50__      | __82.74__ |
> | [1]    | 91.00      | 83.50 | 90.50      | 77.27 |
> | [2]    | 93.00      | 84.00 | 91.00      | 78.82 |
>
> [1]A multitask, multilingual, multimodal evaluation of chatgpt on reasoning, hallucination, and interactivity - Bang, Yejin, et al.
>
> [2]A Preliminary Evaluation of ChatGPT for Zero-shot Dialogue Understanding - Pan, Wenbo, et al
>
> **Necessity of belief state in TOD**
>
> Due to the inability of LLMs to interact extensively with external knowledge, such as querying restaurant availability from a DB, we still need to retrieve keywords from the belief state to perform DB queries. While there may be better approaches in the future, currently, querying external knowledge through the belief state remains a more reliable method.
>
> ***
> __Q4:__ Can you provide some descriptions of the belief state and what is its structure?
>
> __A4:__ The standard format of a belief state is __{domain slot1:value1; slot2:value2; ...}__. However, the improved method used in this paper employs a belief state that does not include values of slots, with the format __{domain slot1; slot2; ...}__. In the actual algorithm, some formatting constraints are applied based on LLMs to reduce generation errors, such as __{domain slot1=?; slot2=?; ...}__.
> ***
> __Q5:__ Are there any ablation studies to investigate the effect of static prompt, dynamic prompt and the prompt for system response to the final results?
>
> __A5:__ The results with static prompts can be referred to in the original paper's 'ours w/o APG' and A3. Below are the additional results from the ablation study of prompts for system responses, and it can be observed that the lack of prompts for system responses leads to a significant decrease in results.
>
> |                                 | Inform | Success | BLEU | Combined score |
> |:--------------------------------:|:------:|:-------:|:----:|:-----:|
> | __Attraction__                       |        |         |      |       |
> | w/o prompts for system responses | 85.00     | 69.00      | 12.90 | 89.90  |
> | __w/ prompts for system responses__  | 98.00     | 87.00      | 4.89 | __97.39__ |
> |----------------------------------|------|------|------|-------|
> | __Hotel__                      |      |      |      |       |
> | w/o prompts for system responses | 74.50 | 43.50 | 8.12 | 67.12 |
> | __w/ prompts for system responses__  | 88.50 | 62.00   | 2.76 | __78.01__ |
> |----------------------------------|------|------|------|-------|
> | __Restaurant__                       |      |      |      |       |
> | w/o prompts for system responses | 81.00   | 55.50 | 12.80 | 81.05 |
> | __w/ prompts for system responses__  | 96.50 | 71.50 | 3.51 | __87.51__ |
> |----------------------------------|-------|-------|------|--------|
> | __Train__                       |       |       |      |        |
> | w/o prompts for system responses | 80.81 | 64.65 | 9.96 | 82.69   |
> | __w/ prompts for system responses__ | 82.74 | 80.71 | 4.98 | __86.71__ |
>
> ***
> __Q6:__ How to measure the format error and dialogue generation error?
>
> __A6:__ We conduct manual analysis of failed dialogues in the test set to determine the reasons behind the failures.
> * If a failure is due to a formatting error that prevents the belief state from being parsed, it is categorized as a 'format error'.
> * If the failure is a result of a system response that does not align with logic, it is categorized as a 'dialogue generation error'.
> * If a belief state is generated that does not match the content, it is categorized as a 'belief state error'.
> ***
> __Q7:__ Another question about this area: This paper introduces an Adaptive Prompt Generation for solving task-oriented dialogue tasks, and it introduces many previous experiences (e.g., Belief state) in TOD tasks. But one question is: after we have LLMs (e.g., ChatGPT), do we still need to follow these traditional settings to solve TOD tasks?
>
> __A7:__ This question is similar to Q3, and you could refer to A3 for a related response.

---

### Official Review · Reviewer_E3rq · 2023-08-04

**Soundness:** 3

**Excitement:**

3: Ambivalent: It has merits (e.g., it reports state-of-the-art results, the idea is nice), but there are key weaknesses (e.g., it describes incremental work), and it can significantly benefit from another round of revision. However, I won't object to accepting it if my co-reviewers champion it.

**Paper Topic And Main Contributions:**

The paper describes a method of dynamic prompting for multi-turn task-oriented dialog generation. Two constituent methods are a Trainable Slot Generator (TSG) - which generates the domain and slot information for the belief state and an Adaptive Prompt Generator for the LLM to generate an appropriate dialogue response.  The framework was evaluated in MultiWOZ2.0 dataset. The results are compared with SOLOIST and ChatGPT methods (for slots and belief state prediction).

**Reasons To Accept:**

1) Novel approach consisting of static and dynamic prompt generation - prompts for belief state and for system response

**Reasons To Reject:**

1) The authors may consider running further experiments especially where slots are complex and there may be more than one belief states

**Reproducibility:**

4: Could mostly reproduce the results, but there may be some variation because of sample variance or minor variations in their interpretation of the protocol or method.

**Reviewer Confidence:**

3: Pretty sure, but there's a chance I missed something. Although I have a good feel for this area in general, I did not carefully check the paper's details, e.g., the math, experimental design, or novelty.

---

> ### Author Rebuttal · Authors · 2023-08-28
>
> __Reviewer#2:__
>
> Thanks for your careful and valuable comments. We will provide an explanation for this point.
> ***
>
> __Q1:__ The authors may consider running further experiments especially where slots are complex and there may be more than one belief states
>
> __A1:__ Due to time and space constraints, we will address this aspect in our future work. We plan to explore more complex scenarios, including multi-domain scenarios or scenarios that involve additional information, such as querying databases.
>
> In addition, we have conducted supplementary experiments on more datasets. For specific details, please refer to A2 to reviewer#3.

---

### Official Review · Reviewer_uoyj · 2023-08-07

**Soundness:** 2

**Excitement:**

3: Ambivalent: It has merits (e.g., it reports state-of-the-art results, the idea is nice), but there are key weaknesses (e.g., it describes incremental work), and it can significantly benefit from another round of revision. However, I won't object to accepting it if my co-reviewers champion it.

**Paper Topic And Main Contributions:**

The paper proposes an adaptive prompt generation framework for task-oriented dialogue systems. The key ideas are:

- A trainable slot generator (TSG) is used to generate domain and slot information from the belief state. This reduces annotation cost and improves stability.
- An adaptive prompt generator (APG) uses the output of TSG to generate prompts for the LLM to produce the full belief state and system response. It maintains query tables to select relevant prompt candidates dynamically.
- Experiments on MultiWOZ show the framework outperforms existing methods in dialogue evaluation metrics. The TSG improves accuracy of slot prediction. The APG mitigates LLM hallucination by constraining prompts.

**Reasons To Accept:**

The method is innovative.
- The idea of using a trainable slot generator to provide interpretable priors for guiding adaptive prompt generation is novel.
- Experiments on the MultiWOZ dataset show the proposed framework outperforms prior SOLOIST and ChatGPT baselines on dialogue evaluation metrics.

**Reasons To Reject:**

* Why not use LLMs for implementing the TSG module also? This reasoning is not available. It's unclear whether TSG is the best option for generating the slots, as the success of the next phase is dependent on this. So, comparison with baselines for TSG should be done more thoroughly.
* Unfair comparison with baseline in response generation task: The SOLOIST baseline is quite old now. Aren't there more recent models proposed for this? Please compare with other baselines from here: https://github.com/budzianowski/multiwoz#combined-score--inform-success05--bleu-1
* Results on generation metrics are not shown. (BLEU, METEOR or newer metrics that can capture semantics e.g. DEB [1])

[1] Improving Dialog Evaluation with a Multi-reference Adversarial Dataset and Large Scale Pretraining - Sai et al.

**Reproducibility:**

3: Could reproduce the results with some difficulty. The settings of parameters are underspecified or subjectively determined; the training/evaluation data are not widely available.

**Reviewer Confidence:**

4: Quite sure. I tried to check the important points carefully. It's unlikely, though conceivable, that I missed something that should affect my ratings.

---

> ### Author Rebuttal · Authors · 2023-08-28
>
> __Reviewer#1:__
>
> Thanks for your careful and valuable comments. We will explain your concerns point by point.
> ***
>
> __Q1:__ Why not use LLMs for implementing the TSG module also? This reasoning is not available. It's unclear whether TSG is the best option for generating the slots, as the success of the next phase is dependent on this. So, comparison with baselines for TSG should be done more thoroughly.
>
> __A1:__ Using black-box LLMs as the TSG module to generate domain and slot information is not very effective.
> As shown in the table below, our proposed TSG outperforms the black-box LLM-based slot generator by 10.6%. It can be seen that black box is difficult to achieve ideal results in complex domain.
>
> We analyze that the reason behind is that the black-box LLMs can not be directly fine-tuned, making them unable to reliably generate the domain and slot information specific to our domain. Additionally, similar phenomena were also observed in other related work [1].
>
> Therefore, they may generate irrelevant slot generation and degrade the performance.
>
> Using white-box LLMs might potentially reduce the amount of data required for fine-tuning. However, due to time and computational limitations, we have not fully investigate this approach. We will further investigate in future work.
>
> [1]A Preliminary Evaluation of ChatGPT for Zero-shot Dialogue Understanding - Pan, Wenbo, et al
>
> |	| Attraction	| Hotel	| Restaurant	| Train |
> | :------- | :-------: |  :-------:  |  :-------:  |  :-------:  |
> | __ours__ | __58.27__	 | __33.02__	 | __55.81__ | 	__63.78__ |
> | ChatGPT |	54.80 | 	30.76 | 	45.31 | 	59.76 |
> | PPTOD	 | 58.11 | 	32.76	 | 54.34 | 	62.90 |
> ***
> __Q2:__ Unfair comparison with baseline in response generation task: The SOLOIST baseline is quite old now. Aren't there more recent models proposed for this? Please compare with other baselines from here: https://github.com/budzianowski/multiwoz#combined-score--inform-success05--bleu-1
>
> __A2:__ We compare with more recent state-of-the-art methods as shown in the table below. We observe from the results that our proposed method significantly outperforms these SOTA methods.
> |  | DST   | Inform | Success | BLEU  | Combined score |
> |:-----------| :-------: | :-------: |  :-------:  |  :-------:  |  :-------:  |
> | _Attraction_ | | | | | |
> | __ours__       | 62.20 | 98.00  | 87.00   | 4.89  | __97.39__ |
> | SOLOIST    | /     | 86.00  | 68.00   | 14.60 | 91.60 |
> | GALAXY     | /     | 92.00  | 62.00   | 9.47  | 86.47 |
> |----------|-------|--------|---------|-------|-------|
> | _Hotel_   | | | | | |
> | __ours__     | 28.90 | 88.50  | 62.00   | 2.76  | __78.01__ |
> | SOLOIST  | /     | 75.00  | 51.50   | 10.09 | 73.34 |
> | GALAXY   | /     | 84.50  | 29.00   | 5.50  | 62.25 |
> |----------|-------|--------|---------|-------|-------|
> | _Restaurant_  | | | | | |
> | __ours__       | 54.90 | 96.50  | 71.50   | 3.51  | __87.51__ |
> | SOLOIST    | /     | 84.00  | 62.50   | 13.17 | 86.42 |
> | GALAXY     | /     | 76.50  | 64.50   | 11.68 | 82.18 |
> |----------|-------|--------|---------|-------|-------|
> | _Train_   | | | | | |
> | ours    | 61.70 | 82.74  | 80.71   | 4.98 | 86.71 |
> | SOLOIST | /    | 80.31  | 74.24   | 11.90 | __89.18__ |
> | GALAXY  | /    | 87.31  | 73.60    | 6.67 | 87.13 |
> ***
> __Q3:__ Results on generation metrics are not shown. (BLEU, METEOR or newer metrics that can capture semantics e.g. DEB )
>
> __A3:__ We include BLEU metrics in the table above. The results show that our proposed method outperforms baseline methods on Combined score. However, LLMs produce a variety of open-ended responses, and metrics like BLEU and METERO cannot fully reflect the model's capabilities e.g.[2], so we excluded it from previous evaluations. We will explore more equitable metrics in our future work, like using LLMs.
>
> [2] USR: An unsupervised and reference free evaluation metric for dialog generation - Mehri S, Eskenazi M.

---

### Meta-Review · Area_Chair_pjVV · 2023-09-11

**Recommendation:** 3

**Metareview:**

This paper introduces an adaptive prompt generation framework designed to enhance the performance of off-the-shelf Large Language Models (LLMs) in task-oriented dialogue systems. The proposed framework incorporates a trainable slot generator that produces domain-specific slots, which are then utilized for subsequent prompt generation. This adaptive prompt generator is capable of generating belief states and system responses. The framework's effectiveness was evaluated using the MultiWOZ 2.0 benchmark and tested in conjunction with SOLOIST and ChatGPT.

The soundness scores were recorded as (2, 3, 2). The reviewers expressed multiple concerns regarding the paper's experimental fairness, highlighting:

1. The use of an outdated off-the-shelf LLM.
2. The absence of generation metrics.
3. The lack of complex-slot tasks.
4. The lack of an ablation study for the proposed sub-techniques.

In response, the authors presented new experimental results during the rebuttal phase, showcasing updated LLMs, new generation metrics, an additional TOD benchmark, and ablation studies. As a result, the reviewers reached a consensus that, if these rebuttal points are incorporated into the final paper, it would substantially strengthen the work.

As for the excitement scores, they were (3, 3, 2). While all reviewers recognized the novelty of the approach, they also noted that some of the sub-techniques were inadequately supported in the initial submission. Consequently, the overall excitement level for the paper is moderate.

---

### Meta-Review · Senior_Area_Chairs · 2023-10-05

**Recommendation:** 3

**Metareview:**

meta review

---

### Decision · Program_Chairs · 2023-10-07

**Decision:**

Accept-Findings

**Comment:**

This paper introduces an adaptive prompt generation framework designed to enhance the performance of off-the-shelf Large Language Models (LLMs) in task-oriented dialogue systems. The proposed framework incorporates a trainable slot generator that produces domain-specific slots, which are then utilized for subsequent prompt generation. This adaptive prompt generator is capable of generating belief states and system responses. The framework's effectiveness was evaluated using the MultiWOZ 2.0 benchmark and tested in conjunction with SOLOIST and ChatGPT.

The soundness scores were recorded as (2, 3, 2). The reviewers expressed multiple concerns regarding the paper's experimental fairness, highlighting:

1. The use of an outdated off-the-shelf LLM.
2. The absence of generation metrics.
3. The lack of complex-slot tasks.
4. The lack of an ablation study for the proposed sub-techniques.

In response, the authors presented new experimental results during the rebuttal phase, showcasing updated LLMs, new generation metrics, an additional TOD benchmark, and ablation studies. As a result, the reviewers reached a consensus that, if these rebuttal points are incorporated into the final paper, it would substantially strengthen the work.

As for the excitement scores, they were (3, 3, 2). While all reviewers recognized the novelty of the approach, they also noted that some of the sub-techniques were inadequately supported in the initial submission. Consequently, the overall excitement level for the paper is moderate.|meta review